

# Wireless laptop-based phonocardiograph and diagnosis

Amy T. Dao

University of New Mexico, Albuquerque, NM, USA

## ABSTRACT

Auscultation is used to evaluate heart health, and can indicate when it's needed to refer a patient to a cardiologist. Advanced phonocardiograph (PCG) signal processing algorithms are developed to assist the physician in the initial diagnosis but they are primarily designed and demonstrated with research quality equipment. Therefore, there is a need to demonstrate the applicability of those techniques with consumer grade instrument. Furthermore, routine monitoring would benefit from a wireless PCG sensor that allows continuous monitoring of cardiac signals of patients in physical activity, e.g., treadmill or weight exercise. In this work, a low-cost portable and wireless healthcare monitoring system based on PCG signal is implemented to validate and evaluate the most advanced algorithms. Off-the-shelf electronics and a notebook PC are used with MATLAB codes to record and analyze PCG signals which are collected with a notebook computer in tethered and wireless mode. Physiological parameters based on the S1 and S2 signals and MATLAB codes are demonstrated. While the prototype is based on MATLAB, the later is not an absolute requirement.

## INTRODUCTION

The electrocardiogram (ECG) is a popular method for checking anomalies of cardiorespiratory function over many decades, and it works by keeping track of electrical heart activity. However, heart defects may be caused by structural abnormalities and therefore are more likely to produce vibromechanical indicators aside from electrical ones. As an example, heart auscultation is more useful than ECG for characterizing murmurs and other abnormal heart sounds. Heart sounds convey important physiological and pathological information (*Kim, Lee & Yeo, 1999*). Heart murmurs caused by turbulent blood flow and anomalous valve opening or closing, can be noticeably detected by trained ears when adequate sensors are used. While auscultation is useful, detection of cardiac signatures via auscultation demands extensive physician's experience, whether with an analog acoustic or electronic stethoscope. It is desirable to equip primary care physicians who do not have extensive auscultation skills with a diagnostic tool so they screen patients for referable conditions. On the other hand, an accurate detection of the cardiac cycle can improve the diagnosis with quantitative details useful for specialists. To meet that goal, many techniques of quantifying the cardiac cycle with improved accuracy have

Corresponding author
Amy T. Dao, daop@comcast.net, daoamina@gmail.com

been explored. Examples of approach include improving detection of the cycle (*Yu et al., 2012*) and reducing of noise (*Wang, Wang & Liu, 2010*). One of the useful cardiac reserve indicators is the diastole to systole ratio that evaluates the adequacy of the volume of blood reaching the heart during diastole. Autonomous detection and classification of cardiac reserve has been proposed (*Liu et al., 2012b*). Inotropic agents belong to a class of drugs that affect the contraction of the heart muscle. At present, ECG is commonly used to test many cardiac agents, however it cannot be used for cardiac inotropic agents (*Liang, Lukkarinen & Hartimo, 1997*). Long term monitoring of the mentioned cardiac indicators may be more accessible with the use of a wireless and portable PCG system. It may also be beneficial for general users, patients and front line care givers to perform auscultation at home and to continuously monitor sporadic symptoms that may not be detected during periodical medical visits. In other words, patients can collect persistent long term data for the physicians. Furthermore, the convenience of a sensor not tethered to the recording PC allows continuous monitoring the patient in many relevant scenarios, such as treadmill or weight lifting exercises. Therefore, an automated and wireless system to detect and characterize heart sounds is explored in this paper. Variance of PCG quality, whether due to electronic specifications of the sensor, the placement of the stethoscope on the chest and additional noise introduced by the wireless operation are seen as major challenges on the sensor side. On the signal processing side, we would like to show that the advanced PCG algorithms reported in the literature can be implemented on a modest computing platform. The goal of the paper is to report the implementation of a simple wireless PCG sensor designed to operate with a notebook or tablet computer, and the value of signal processing in minimizing the effects of the varying electronic performance, ambient noise and stethoscope's placement. The group of users targeted by this sensor consists of primary care physicians and care givers. Therefore, key requirements are robustness of the processing algorithms, immunity to the mentioned variances, informative indicators and a rudimentary classification of heart sounds to assist users in choosing the next action.

An essential function of the PCG signal processing is the extraction of the first (S1) and second heart sound (S2). A survey of heart sound segmentation techniques based on the extraction of the waveform envelope was conducted by *Choi & Jiang (2008)*. The paper evaluated the extraction techniques which are based on the Shannon energy envelope, Hilbert transform waveform, and characteristic waveform. A more recent evaluation of envelope extraction algorithms was reported by *Liu et al. (2012a)*. We tested the use of a novel technique developed and reported by *Barabasa, Jafari & Plumbley (2012)* that has been proven to be insensitive to performance degradation and noise interference, a potential major issue for wireless sensors and recording during physical activity. This algorithm is also robust with respect to pathological signals such as heart murmurs. It is based on musical analysis applications, and particularly known for its ability to track beats in the presence of noisy and varying background. We adopted the particular technique of dynamic programming for beat tracking published by *Ellis (2007)*. Robust segmentation of the heart sounds is only the first step in classifying heart sounds. It has been proposed that diagnostic parameters (*Choi & Jiang, 2005*), derived from the heart sounds and cardiac

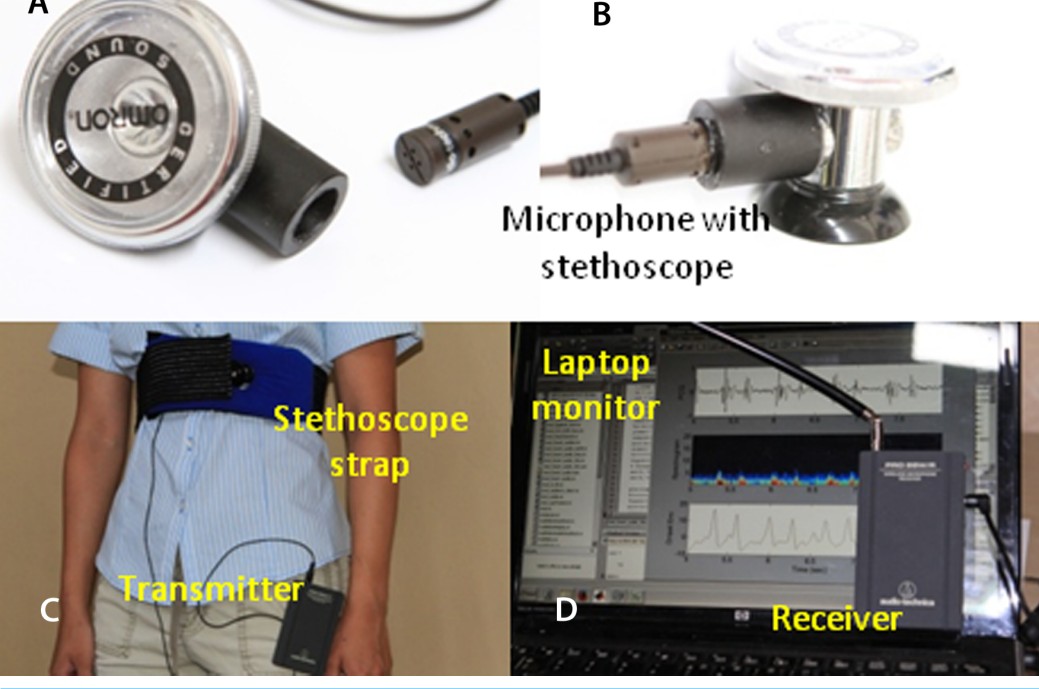

**Figure 1 Experimental setup.** Off-the-shelf microphone and stethoscope (A). Side view of the assembled stethoscope head (B). (C) the stethoscope strap; (D) laptop's screen and microphone receiver unit.

waveform, can be used for classification and monitoring trends. Our goal is to demonstrate that useful physiological parameters can be derived from heart sounds and presented to care givers for screening purposes.

Many medical algorithm development works are reported without implementation details. That makes it difficult to estimate the effort requires to transition research knowledge to commercial realization. In this paper, we will make an effort to trace the lineage of the open source codes, describe the modifications in sufficient detail to aid the readers in reproducing results and duplicating the prototype. While the sensor we built is not optimum for mass production, there will be sufficient technical specifications for anyone interested in such an endeavor.

## SYSTEM AND PROTOTYPE HARDWARE

The wireless microphone system is based on the commercially available Audio-Technica Model number ATR288W (Audio-Technica, Tokyo Japan; $131.00). Wireless communications between the transmitter unit and the receiver unit are established via 2 VHF channels: 169.505 MHz and 170.305 MHz. To improve performance, we purchased and used a Lavalier condenser microphone (Audio-Technica AT829MW; $37.00) to replace the microphone that came with the ATR288W. The microphone is coupled to the stethoscope (Omron Sprague Rappaport; Omron, Kyoto, Japan; $17.00), as shown in Figs. 1A and 1B, and connected to the transmitter which can be worn by the subject (Fig. 1C). The receiver's output is connected to the MICROPHONE input of the laptop. The maximum

sampling rate of 44.1 kHz and amplitude resolution of 16 bit were selected via software control and typically used in this project. The PCG software determines the sampling rate according to the purpose of the run. The frequency response window from 35 Hz to 20 kHz is sufficiently wide for PCG waveforms. Low-pass filtering implemented in software is used to control the upper frequency limit to 1000 Hz. The ATR288W is compatible with both Macintosh Mac OSX and Windows XP, Vista, 7 and 8 (USB 1 and 2). This compatibility allows choosing any computer platforms from tablet to notebook size.

A chest strap was made from a body icing kit purchased from CVS pharmacy (Caldera Multi-Purpose Therapy Wrap; SCO, Lindon, Utah, USA; $12.99). The kit was modified after the gel was removed. Polyester foam ($5), sold for pillow stuffing, is inserted into the pad sleeve to shield the microphone from acoustic noise and to provide a cushioned contact with the chest. A hole in the pad allows positioning the microphone in the middle of the pad and keeping it in contact with the chest (see Fig. 1C).

Any computer with a MICROPHONE input will work for this application. Our prototype is a notebook PC running Windows 7. While MATLAB computing language is not required in general, for rapid prototyping and easy leveraging of research algorithms available in the public domain, MATLAB R2013b, a scientific and engineering computing framework produced by Mathworks, is used to write the program. Figure 2 shows a raw wired PCG waveform and a raw wireless PCG waveform. It is apparent that the signal to noise ratio of the wireless signal is comparable to that of the wired signal. The most challenging aspects of wireless PCG recording is to keep the stethescope stationary when the subject jogs or walks on a treadmill. In this situation, additional noise can be picked up by the microphone or the strap may shift enough to affect the signal strength. Fortunately, most of the adverse effects are alleviated by the use of advanced segmentation techniques.

## SEGMENTATION TECHNIQUES

The detection of the heart sounds S1 and S2 is accomplished with a beat finding technique developed for the music industry as discussed in Barabasa's paper (*Barabasa, Jafari & Plumbley, 2012*). The specific beat tracking technique is based on dynamic programming (*Davies & Plumbley, 2007*). In the first step of the detection algorithm, audio signal is converted to its onset strength envelope (*ose*). The *ose* is calculated as the sum of the difference between the spectra of the current and the previous waveform segments. The *ose* therefore represents the instantaneous overall change in spectral content (distribution of energy at different frequencies). To calculate the *ose*, a window of N data points is advanced in equal steps until the window reaches the end of the waveform. The number of data points N in each window

$$N \cong F_S/8 \tag{1}$$

corresponds to 1/8 s for the selected audio sampling frequency. The step is only half the size of the window so there is overlap between consecutive windows. The window is analyzed to calculate the spectral content or the energy contained in 20 frequency bins. The *ose* is
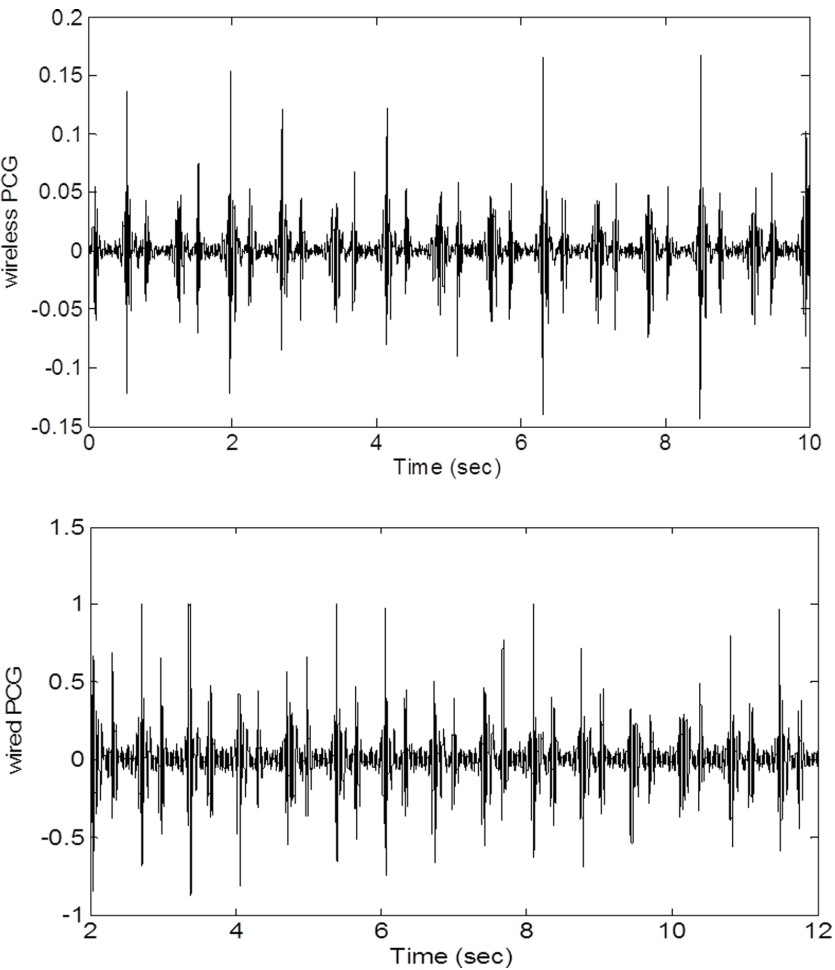

**Figure 2 Wired and wireless signals.** A comparison of wired and wireless amplitudes shows that the voltage of the wireless signal is lower but the signal-to-noise ratios (quality) are comparable.

calculated at each step $k$ as follows.

$$\Gamma(k) = \sum_{m=1}^{20} |S_m(k) - S_m(k-1)|^2. \tag{2}$$

The differences in power ($S_m$) in each of the 20 frequency bins between step $k-1$ and step $k$ are squared and summed. The expression assumes that the *ose* correlates with the occurrence of a beat. As such, the likelihood of a beat is proportional to the magnitude of the change in spectral content and not to the amplitude of the waveform itself. Figure 3 shows the PCG waveform (Fig. 3A), the spectrogram (Fig. 3B), where the energy in each spectral band (frequency) is represented by color shading and the *ose* (Fig. 3C) for the same time window. Note that the strength of the onset envelope is highest when the spectral contents begin to change. Other techniques of envelope extraction determine the beat as the time the waveform's amplitude or energy exceeds a threshold, hence placing the beat at a time slightly later than the one predicted by the *ose*. The MATLAB script *beat.m*

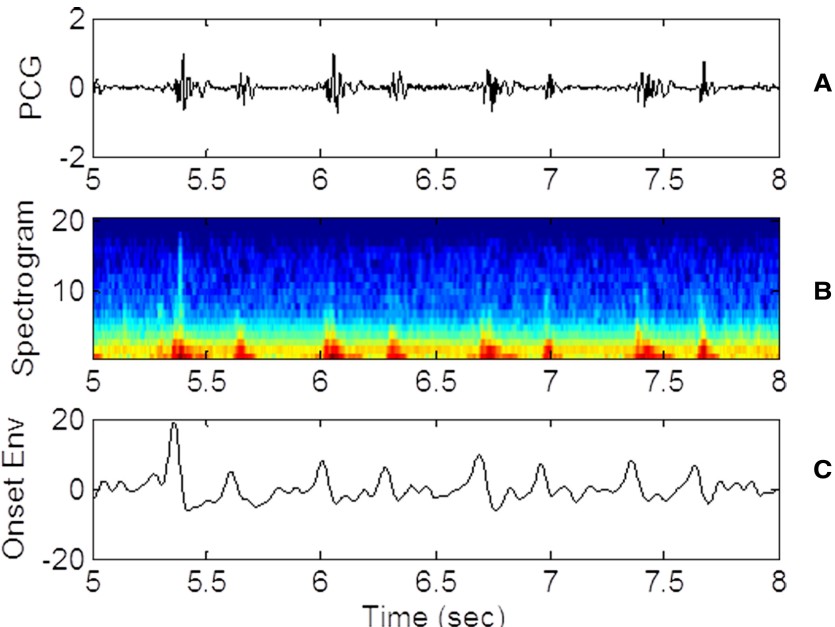

**Figure 3 PCG, spectrogram and *ose* waveforms.** (A) 3 seconds of raw PCG record showing voltage vs. time. (B) corresponding spectrogram vs. time. (C) derived onset strength envelope. Color represents energy contained in each band (black, lowest; and dark red, highest). The spectrogram indicates that PCG wave energy is concentrated in the low frequency bands as expected. Energy is also concentrated at the times of the heart sounds. The *ose* reflects the total change in band energies and coincides with the onset of the "high energy" regions.

and all supporting functions which are distributed as open source codes (*Ellis, 2014*) are incorporated in our codes. The *beat.m* algorithm also encourages conformance to a global tempo which was pre-computed for the entire record. The use of the *ose* and conformance to a global tempo improve the technique's robustness and immunity with respect to ambient noise.

The beat tracking algorithm is applied to sequentially detect the two sequences of heart sounds, S1 or S2. After the first sequence of beats is detected, its signature needs to be removed before the beat tracking algorithm is applied the second time to find the second sequence. The removal of the signature of the first sequence is accomplished by multiplying the original *ose* waveform with a weighting function. The weighting function is defined as a constant of unity everywhere except near the times of the first sequence of beats. Near those times, the weighting function is set to a small value find the following form quite effective.

$$W(t) = 1 - \sum_i 0.8 \cdot \exp(-(t - \mu_i)^2/2\sigma^2), \quad i = 1 \ldots N_b \tag{3}$$

where $t$ is time, $i$ the beat index, $N_b$ the number of detected beats, $\mu_i$ is the time of the ith detected beat (in the first sequence) and $\sigma$ the temporal width of the troughs in the weighting function. Figure 4 shows the original *ose* (Fig. 4A), the weighting function (Fig. 4B) and the processed *ose* (Fig. 4C). The original waveform displays prominently the two interleaving sequences of heart beats. Applying the tracking algorithm the first

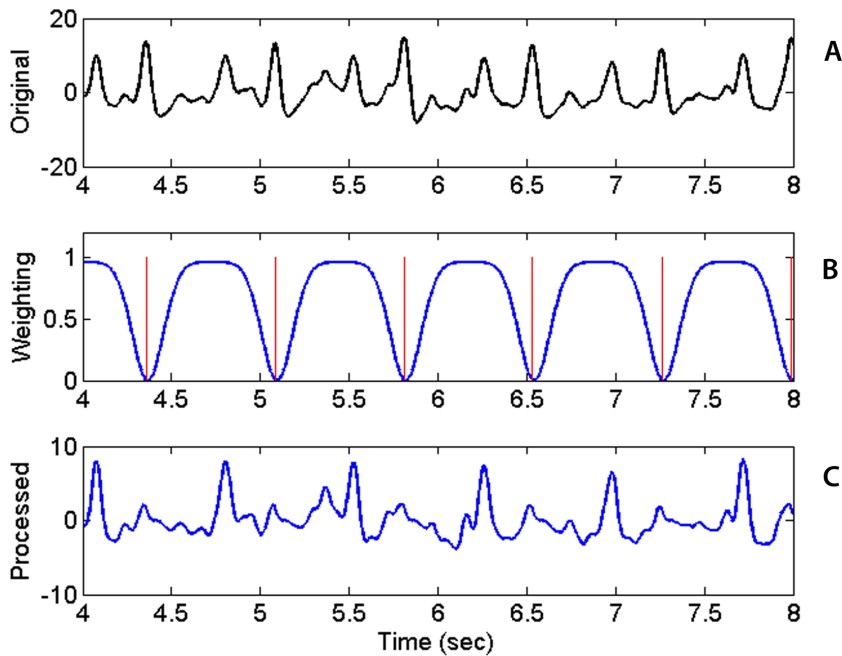

**Figure 4 Raw waveform, weighting function and processed waveform.** Upper panel: Original Onset Strength Envelop (*ose*) waveform as a function of time. Middle: Weighting Factor waveform with locations of detected beats marked by vertical red lines. Bottom: Processed *ose* waveform showing previously found beats practically removed after the multiplication with the weighting factor.

time detects the sequence of stronger heart sounds which happens to be S2 in this case and as shown in Fig. 4. The locations of the troughs are chosen to coincide with the already detected beats and marked with the red vertical lines in Fig. 4B. The product the original *ose* and the weighting function produces a new waveform (Fig. 4C) in which the signature of the first sequence of beats has been dampened and practically eliminated. With the first sequence eliminated, the algorithm is applied once more to retrieve the second sequence of beats. With both sequences retrieved, one still has to identify which one is S1 because the original *beat.m* algorithm cannot distinguish one from the other. Our codes identify the S1 sequence by inspecting the timing relationship between consecutive beats in the two sequences and the spectral content in the interval between the two heart sounds. Specifically, the separation between consecutive S1 beats cannot be less than 0.22 s or greater than 1.3 times the average heartbeat interval of that collect. The fact that the waveform segment that begins with S1 and ends with S2 always contains higher infrasonic-frequency variance is used to differentiate S1 from S2. The sequence of beats that satisfies those conditions are identified as S1.

## DATA COLLECTION ROUTINE

Data collection starts first with strapping the microphone over the heart of the examinee, secondly the examiner putting on the headphones to monitor the recording and to ensure that the signal strength is sufficiently high but not too close to saturation level, and thirdly the examiner commanding the MATLAB program to record heart sounds and display

the PCG signal. A frequently used record length of 50 s, recording 55 to 100 heartbeats, is sufficiently long to warrant that the timings of the first and second heart sounds are statistically significant for a relatively constant heart rate or when the subject is at rest. Sometimes, records of 200 s or longer are collected to study the change of heart rate in the recovery phase after physical exercise. In those cases, the objective is to monitor the gradual decrease of heart rate in the recovery phase. In this proof-of-concept study, the PCG signal was recorded to show that useful physiological indicators can be acquired. The study is not intended to validate the tool's clinical readiness. With the intended scope, the numbers of subjects (five) and samples (26) are deemed sufficient. Since the objective is only to capture the timing of the S1 and S2 sequences and not to diagnose particular aspects of the hemodynamic response, auscultation placement is straightforward and doesn't require cardiologist's expertise. For our purpose, placing the stethoscope near the heart's apex typically results in a strong signal to noise ratio which is the most important factor in capturing the heartbeat sequence timings. The stethoscope microphone is connected to the transmitter unit and the receiver is connected to the laptop to record heart sounds. A pair of headphones is also connected to another port in the laptop configured to monitor the audio. Ideally, the microphone only senses the heart sounds of the subject and not ambient noise. Thus, data collection is best in a quiet room, with the subject sitting completely still, and the chest strap adjusted so that the microphone is directly over the heart. However, the processing techniques we use are effective in alleviating the effects of extraneous noises. When needed, the subject may wear the wireless microphone and jog on a treadmill while data is being collected. The data taker, listening through the headphones, can help with the adjustment of the microphone gain and placement of the sensor over the heart.

## ANALYSES AND RESULTS

In a typical data collect, 50 s of audio data are collected using the MATLAB *audiorecorder* built-in function, at a rate 32,000 samples per second. The entire record consists of 1,600,000 values. The block diagram of the codes is shown in Fig. 5 for reference. Since the sampling rate is much higher that the highest frequency found in actual heart sounds, signal with frequency higher than 1000 Hz is filtered out. The beat tracking script, *beat.m*, made available at the LabROSA internet site (*Ellis, 2007*) was designed to extract a single dominant beat, not two beat sequences as in the case of heart sounds. We modified the codes to extract both heart sounds by running the algorithm in two passes. After the first pass, the signal that corresponds to the first detected sequence of heart sounds is removed and the pruned signal is processed again to detect the second sequence, as described in 'Segmentation Techniques.'

Using the timing relationship between the S1 and S2 sounds, we proceeded to identify S1. The S1 and S2 beats are subsequently paired up and the beat intervals (T11) and the systolic intervals (T12) are calculated as shown in Fig. 6. The beats which are not detected because of noise and their potentially unpaired beats are not analyzed. We will discuss how this mode of operation contributes to the robustness of the algorithm in 'Discussion' and 'Conclusions.' Note that the instantaneous heart rate can be estimated in

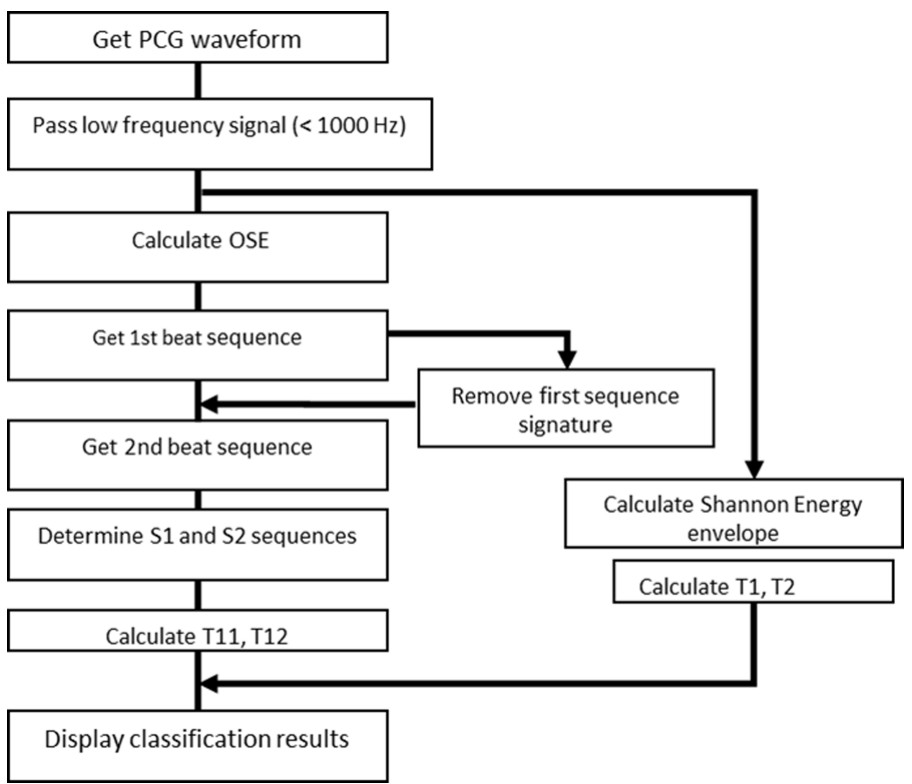

**Figure 5** Block diagram of PCG program logic.

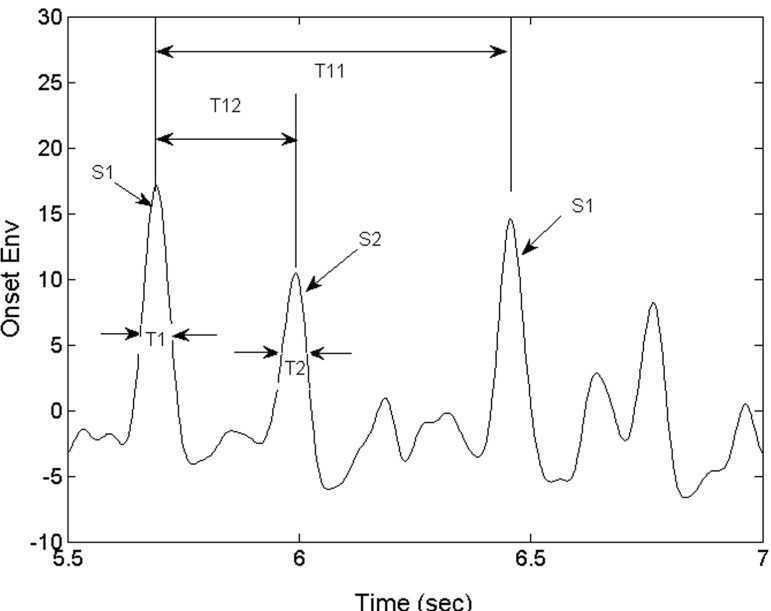

**Figure 6 Derived parameters of the heart sounds.** S1 and S2 are the instants of the first and second heart sound. T11 is the heart beat interval. T12 is the interval between the first and second heart sound, or systole. T1 and T2 are the temporal widths of the first and second heart sounds.

real time by calculating the inverse of T11. Following (*Choi & Jiang, 2005*), two additional diagnostic parameters, heart sound temporal width T1 and T2 (Fig. 6), are calculated directly from the Shannon energy envelope (*see*). Note that they are not derived from the *ose*. The heart sound is composed of several frequencies, all measurable by the PCG and should be included in the *see* though not all are within the human audio spectrum. The *see* which is calculated from acoustic energy in all frequencies may be different from the humanly perceived heart sound. We would like to hypothesize that the *see* is an unbiased representation of the mechanical sound. Therefore, T1 and T2 extracted from the *see* envelope are representative of the mechanical sound made by the heart. The program displays the four diagnostic parameters and indicates the range of nominal values. These physiological parameters are useful for primary care physicians in screening referable patients and for specialists to infer preliminary diagnosis. It's conceivable that the primary care physician may select to send forward the information generated by this system to the specialist prior to the referred visit.

## DIAGNOSTIC PARAMETERS

The physiological parameters of interest consist of the instants of the first heart sounds, S1 and S2 and the timing parameters derived from them. It is conventional to define the characteristic times as in *Choi & Jiang (2005)*. The interval T11 between consecutive S1 occurrences, or heartbeat interval, is defined as shown in Fig. 6. Also shown in Fig. 6, are T1 and T2 -the temporal widths of S1 and S2.

Determining S1 directly with the raw PCG waveform is difficult because the sound consists of a number of modulations. S1 is typically determined based on an envelope waveform that represents the heart sound. While the exact time of S1 depends on the technique of segmentation, the inter-beat interval is less affected by any bias on S1 itself. As pointed out previously, the heart sound instant retrieved by our segmentation technique is biased towards the onset of the sound as opposed to the time when the sound exceeds an arbitrarily chosen threshold. Our technique is therefore not subject to timing bias related to the arbitrary choice of the threshold. Our S1 times are also slightly ahead of the ones chosen by other segmentation techniques. The systolic period (T12), the interval between S1 and S2, is as shown in Fig. 6. The diastolic period, the interval between the current S2 and S1 of the following heartbeat, is calculated as $T21 = T11 - T12$. Note that T12 and T21 are in principles not affected the mentioned bias. As an example of its usefulness, the relationship between the instantaneous heart rate (1/T11) and the systolic and diastolic periods, T12 and T21, was reported to be a useful indicator for patients who are resting, exercising or taking medication (*Bombardini et al., 2008*). Detection of cardiac cycle anomalies in patients with deficiency in cardiac filling, shown as an elongation of the systole and a shortening of the diastole, is another example of its use. A reversal of the systolic/diastolic period ratio, e.g., increasing from less than 1 to above 1, may indicate a compromised cardiac function, e.g., a deficiency in cardiac filling.

Several useful indicators are represented by (a) the systolic and diastolic durations and (b) how these parameters vary with heart rate (1/T11). Not only does exercise
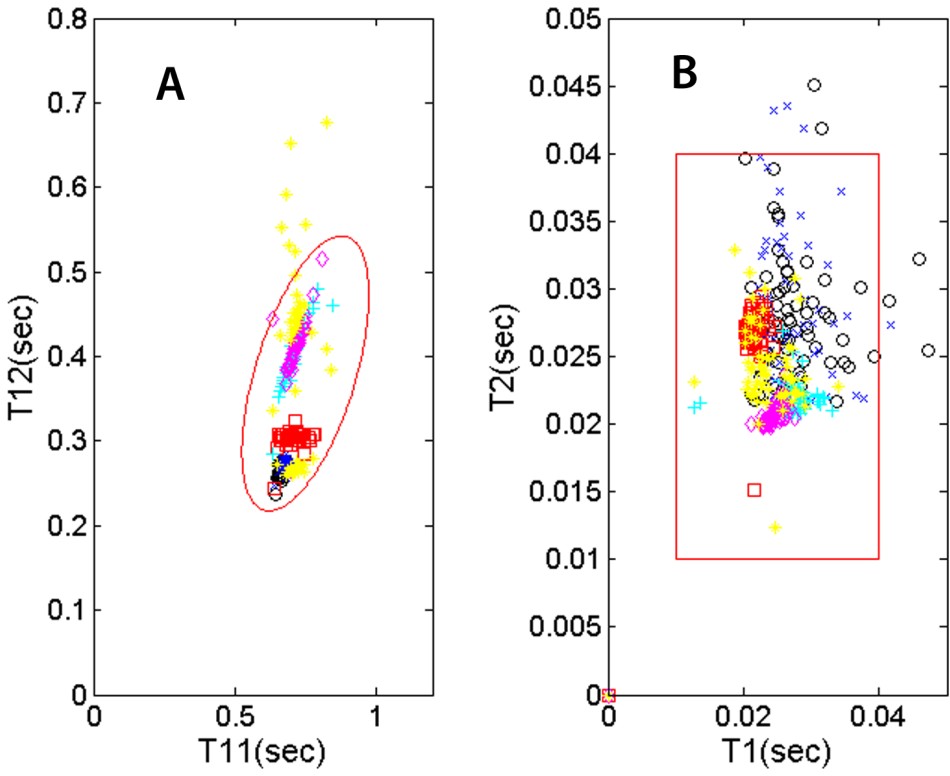

**Figure 7 PCG physiological parameters.** (A) systolic duration T12 vs. heartbeat interval T11. (B) S2 temporal width vs. S1 width. The symbol and color legend is described in the text.

accentuates systolic-diastolic change but in the recovery, patients with heart conditions or on medication may show a recovery trend different from that of a normal person. While this study does not assume any knowledge of the subjects' health conditions, we'd like to present a number of physiological parameters that may be useful for monitoring the mentioned trends. The locations of the T12-vs.-T11 data points on the plot (Fig. 7A) vary from individual to individual. For a given individual, the location will also vary with heart rate. This type of variability can be monitored with the examinee jogging/walking on a treadmill or recovering from physical activity. The plot in Fig. 7A, showing T12 systolic duration plotted against heartbeat interval T11, displays the mentioned types of variability. Six recordings of five individuals are shown in the plot. The legend is as follows: 50 sec recording of subject 1 as black circles, 50 sec for subject 2 as blue crosses, 5 min for subject 2 on treadmill as red squares, 5 min for subject 3 recovering from light exercise as cyan pluses, 5 min of subject 4 recovering from moderate exercise as magenta diamonds and 2 min of subject 5 as yellow pluses. The nominal ranges of the parameters are shown as the tilted ellipse. Only 0.4 % of the data points reside outside of the nominal range. This small percentage may indicate that there is practically no contribution from noise signatures erroneously recorded as cardiac signatures. The shown nominal range is not intended to be the range for normal or healthy subjects and it's beyond the scope of this study to determine the range for normal people. However, it is hypothesized that the locations

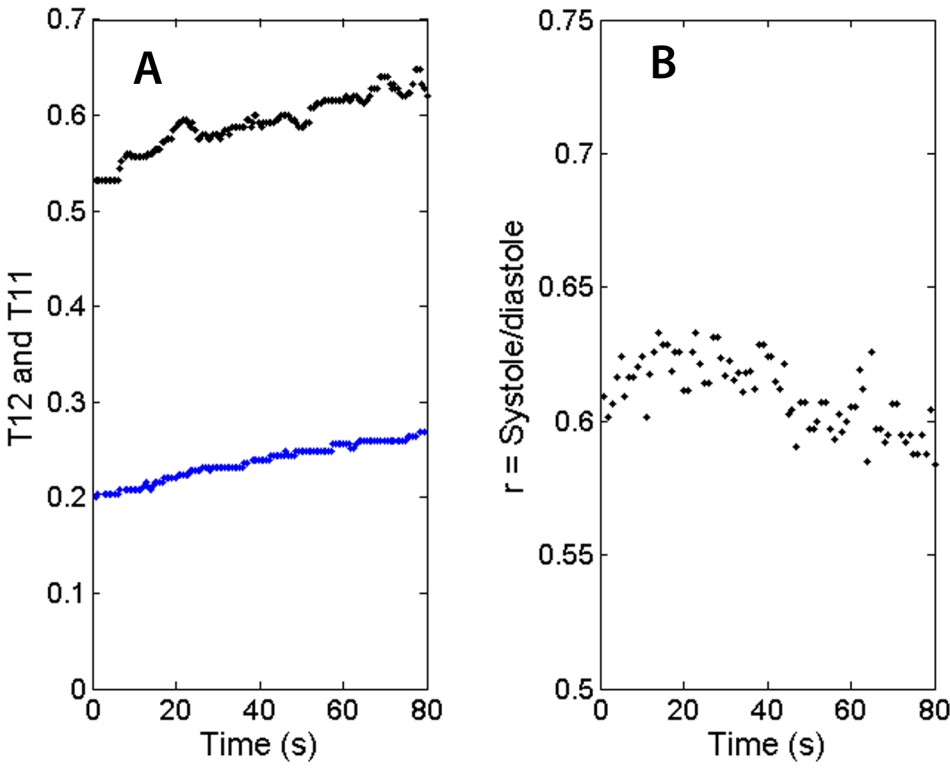

**Figure 8 Trends of physiological parameters.** (A) Systolic durations (T12 as blue dots) and heartbeat intervals (T11 as black dots) in recovery phase. As the heart rate slows down, T12 and T11 recover at slightly different rates. (B) The ratio of systolic duration and diastolic duration shows a slight downward trend for this subject.

of T12-vs.-T11 points and their trends may contain useful physiological information. Similarly, the plot in Fig. 7B shows the variability of the S1 and S2 temporal widths, or T1 and T2. The data points are shown in different colors and symbols according the previously described legend. The large square indicates the region where the T2-vs.-T1 data point would fall for this group of subjects. To calculate the widths, we did not use the *ose* but used the *see* envelope instead (*Choi & Jiang, 2008*).

Because the segmentation and detection of heart sounds is based on a novel beat-tracking technique used in music research, it is inherently more immune to ambient noise and occasional missing beats. The segmentation technique is also robust with respect to varying heart rate. Together with the ability to operate wirelessly, these attributes are essential for PCG recording when the subject is walking, jogging or recovering after physical exercise. Fig. 8A shows the trends of the heart beat interval (T11) and systolic duration (T12) in the recovery phase. Both the heartbeat interval T11 and the systolic duration T12 show a gradual increase as the heart rate slows down. Figure 8B shows the recovery of the systolic/diastolic period ratio as the treadmill slows down. The ratio of systole over diastole, defined as follows, is plotted as a function of time.

$$r = systole/diastole = T12/(T11 - T12). \qquad (4)$$

For this individual, the ratio which is never higher than 0.8 would be considered normal according to *Bombardini et al. (2008)*. It's worth noting that as the exercise winds down, the ratio r slightly decreases, indicating a recovery in cardiac filling efficiency. Again, this study does not assume knowledge of the subjects' health conditions, but the subject in this measurement is a 23-year-old regular jogger.

In the PCG measurements, we found untethered wireless PCG a convenient tool for treadmill measurements and the noise due to treadmill jogging/walking not critically affecting the recording. Even when the interfering noise makes the algorithm miss a few beats, the general tempo was still observed and the recording of the rest of the characteristic times unaffected. The sensor can record the diagnostic parameters from the beginning of the exercise to the end of the recovery phase.

## DISCUSSION

The cost of material is $203 and the cost of the programming software (Matlab Student's version) is $49, although the software has been bought for previous work. The total cost is well within the limits of a typical student research project. The performance is evaluated based the ability to detect all of the beats for the first and second heart sounds. We use the success rate as the metric of performance. The success rate is calculated as the ratio of the number of detected beats over the total number of beats. The latter can be readily determined using the average heart beat interval, a reliable product of the beat tracking algorithm. Since there are no independent measurements of the heart beats, the success rate can only be estimated as mentioned. When the microphone's volume is properly adjusted, the success rate is better than $97+/-2\%$ when the individual is in at rest and $92+/-3\%$ when he/she jogs on a treadmill. To check the validity of our estimates, we also confirmed the success rate by manually inspecting four 50-second records. Those manual determinations of the rate confirmed that the rate is better than 95% in that small sample. Note that the success rate has no bearing on the accuracy of the T11, T12, T1 and T2 values which are based on the detected beats. The missing beats were ignored.

The advanced segmentation technique, based on beat tracking algorithms developed for the music industry, relying on change in frequency contents instead of change in energy, has been instrumental in making the algorithm robust and immune to variation in background noise, heart sound volume and heart rate. It can also be argued that the beat-tracking *ose* is suitable for determining the timings of S1 and S2 because the onset of an acoustic event tracks the rhythm of the events more faithfully than loudness. That is certainly true when noise, sometimes louder than the heart sound itself, is present. Although segmentation of the S1 and S2 sounds is achieved by detecting frequency content change, the width of the heart sounds is obtained using the Shannon energy envelope. One of the reasons to use the *see* waveform to calculate the S1 and S2 temporal widths is so that they can be compared with previous benchmarks. A more important reason is that the *see*, an indicator of mechanical power, has the potential of representing the heart sound with better fidelity than any techniques that rely on variance in a range of frequencies, including human audibility.

To monitor patients conducting physical activity or recovering from it, it is best to have a PCG sensor and analysis techniques which are immune to ambient noise and physiological variability. The technique we implemented is found to retrieve the heart sounds reliably under these strenuous conditions with a success rate better than $92+/-3\%$. The sensor is a prototype system capable of producing useful physiological parameters. The first and second heart sounds, as well as additional "diagnostic" parameters, T1, T2, T11, and T12, could be recorded reliably and displayed in plots that convey pathological information about the cardiac cycle. In Figs. 7 and 8, we proposed specific formats to present these indicators. They are shown relative to a proposed range of normalcy. The proposed range has not been validated by rigorous medical studies and should only be viewed as reference points in this concept of operation.

## CONCLUSIONS

The objective of demonstrating that a low-budget wireless PCG recorder and analyzer can achieve satisfactory performance with modern analysis techniques is met. The performance and the effectiveness of this wireless PCG as a medical tool cannot be evaluated and validated within the scope of this study. Such a study would involve specialists that can evaluate the complimentary utility provided by this screening tool when it is used as a sentry for more standard cardiac diagnostic tools. In such a study, an understanding of the likelihood of false negatives and positives would be required. However, it is shown with this prototype that relevant physiological parameters can still be retrieved and presented to the users (e.g., primary care physicians). We hope that this proof-of-concept paper stimulates interest in developing cost-effective and accessible tools for the front line physician who is responsible for screening referable cases. We foresee wireless PCG equally useful in a non-clinical environment: patients needing long term and persistent monitoring in a home care setting with or without the assistance of care providers. In this case, its main purpose is to provide warning indicators and trends which are made accessible by persistent data collection. In the future, we would like to extend the study to include anomalous and pathological heart sounds to assess its clinical effectiveness.

## ACKNOWLEDGEMENTS

The faculty in the Chemistry Department of Amherst College has been instrumental in this project and extremely helpful with their advice and encouragement in the first phase of this project. We benefited from the Lab for Recognition and Organization of Speech and Audio-*coversongID* software which was generously distributed by Professor Dan Ellis, University of Columbia.

### Funding

The author declares there was no funding for this work.

## Competing Interests

The author declares that there are no competing interests.

## Author Contributions

- Amy T. Dao conceived and designed the experiments, performed the experiments, analyzed the data, contributed reagents/materials/analysis tools, wrote the paper, prepared figures and/or tables, reviewed drafts of the paper.

## Supplemental Information

Supplemental information for this article can be found online at http://dx.doi.org/10.7717/peerj.1178#supplemental-information.

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
