# Peer review of "Wireless laptop-based phonocardiograph and diagnosis"

_PeerJ, doi:10.7717/peerj.1178_

## Round 0.1 · original submission · Major Revisions

Your paper was considered to be a potentially important "proof of concent" paper. Therefore, you should consider resubmitting your paper after appropriate attention to the reviewers comments.

·

Basic reporting

The paper describe a proof-of-concept for a phonocardiographic device and it could be considered a suitable unit of publication.
The paper requires a mother-tongue proof reading.
The background and introduction sections are well covered as well as appropriated bibliographic references are reported.
The structure of the article is not completely conform to the structure as suggested in “Instructions for authors”; however, the sections of the paper seem to be adequate to present a proof-of-concept work, as is the case of the proposed paper.
All figures are clear and well labelled.

Experimental design

The topic of the paper covers the Aims & Scope of PeerJ.
However, the originality of the research is rather questionable, mainly for three reasons.
Firstly, devices very similar to the device described in the paper are already available on the market, both for professional and consumer targets.
Secondly, the hardware was simply obtained from the assembling of different commercial products.
Thirdly, software algorithms were already used by other authors in similar or complementary scientific fields.
However, I recognize that the research question is quite intriguing, i.e. the availability of PCG information in general environment and not only in ambulatories/hospitals.
Moreover, all methods were clearly described by the author and the investigation seems to be conducted with accurate technical standards.

Validity of the findings

All presented data are acquired from only few (one/two?) subjects.
Presumably, only healthy subjects were enrolled in the study.
No statistical analysis was conducted on acquired data.
This is in line with the characteristic of proof-of-concept of the paper.
For this reason, some conclusions about the high performances and the high quality diagnostic capabilities the system could offer are quite speculative.
In particular, the potentials of the signal processing algorithm, when it is used in highly noisy environment, are only extrapolated from already published benchmarks.

Additional comments

The paper address a very important topic, as the use of PCG signal processing in general environment and not only in ambulatories/hospitals.
The characteristic of the electronic device and the implementation of the software algorithms are correctly and widely described.
However, only few data are presented, thus keeping the work at the level of proof-of-concept.
For this reason, some conclusions about the diagnostic capabilities of the system and its robustness against noise are speculative and not fully supported by data.
We strongly suggest to acquire data from an adequate numbers of both healthy and pathological subjects in order to construct a robust statistical analysis.

·

Basic reporting

Figure 6 does not have legends and is not very clear and confusing.
Maybe adding a legend would help. Spliting the Figure into two (one with T12 vs T11 and another with T2 vs T1) would increase clarity, since the author is conveying two different informations.

Experimental design

In Section 3, the paragraph from lines134 to 140 there are missing symbols (line 134), and the explanation is quite convoluted. Maybe expanding would help. Another way of making things more clear would be using figures.

Line: 114: I think the word 'windows' is missing.
In Line 144: why there are two recording durations? It is not clear why the author choose to have 50 seconds or 200 seconds. It is also not clear what data is used in the analysis futher in the article.
In Line 145/146: it is stated that the recording of PCG is only for demonstration and not intended as a study of human subjects. However, the article uses Signal processing and algorithms for heart sound analysis in humans (the whole article use humans!). Even the first phrase in the abstract implies human diagnosis. This phrase does not seems to fit the rest of the article. Maybe the author means that this prototype is not ready clinical use in humans?
in Line 150/151: I am not sure if the user is the best person to jugde the quality of the sound: different auscultation places will allow you to diagnose different diseases (and hear different aspects of the heart's Haemodynamic system). The user may not be the best person to judge the quality of the sound. It would be interesting to validate this with a cardiologist.
In Line 168/169: I don't understand what the author means with the last phrase: "Users can use the display to classify the cardiac function..." Is the author implying that the user can carry on diagnosis based on automated analysis? Not sure if the intention here is to use the display as a measure of signal quality (what is the user seeing here? the waveform? Its envelop? the values of T1,T2,T11,T12?) or if the author is saying that the user can classify his/her own heart sound using author's invention. The latter is usally very dangerous, since it borderlines autodiagnosis and should preferably be avoided.
In Line 170: This section actually talks about features that can be used (by a cardiologist) to infer diagnosis. These features are actually physiological features, and usefull for diagnosis. The article's abstract beautifully mentions that the PCG signal processing algorithms “are developed to assist the physician...”, this section is doing exactly that: using signal processing in the PCT to give physiological measures, so the physician can perform his/her diagnosis or refer the patient to a specialist. This item is merely a suggestion for improvement and is optional.
In Line 181/182: Cardiac cycles anomalies can have different influences in systole and diastole: the text seems to suggest that there is only one type of variation.
In Line 182/183: It would add to the text to use the medical name of this “deficiency in cardiac filling”.
In Line 194: Figure 5 seems to be better to show the widths of T1 and T2.
In Line 204: Is the formula in line 201 existent in the literature or was it created by the author? If it exists in the literature, it should be referenced. If not, some explanation would be good.

Validity of the findings

Lines 211/212: Throughout the paper, there is no mention about the budget used to build the wireless PCG recorder. It is also not mention what the author means by 'performance': execution time? Specificity? Sensitivity? There is no mention of any of these quantities in the paper. This should be added to the paper or these claims (low-budget and high performance) must be removed.

Additional comments

This is a paper about the development of a wirelless Phonocardiograph with physiological signal extraction algorithms, where in addition to the development of the hardware, the author also used an advanced signal processing algorithms to extract some physiologically meaningfull features usefull to the screening of heart diseases.
The article is novel in the hardware setup, since it creatively uses off the shelf components to create such system.
On the Signal processing side, it seems to use an already published heart sound segmentation algorithm and does some analysis of the data, presumably in normal individuals. However, there is no information about the number of individuals used, nor the length of the auscultations. It is also not stated if a cardiologist has validated these heart sounds as normal or not.
It is a good frist phase project. The engineering part of the paper is sound and the analysis of the algorithms seems correct. The results are interesting. There is a need of some medical knowledge when interpreting some of the data. Maybe it would be interesting to ask a cardiologist to read the paper and review the methodology/results.

---

## Round 0.2 · accepted · Accept

I am happy to accept this paper as it stands.